# ClpAP proteolysis does not require rotation of the ClpA unfoldase relative to ClpP

Sora Kim[1†], Kristin L Zuromski[2†], Tristan A Bell[1†], Robert T Sauer[1], Tania A Baker[1]*

[1]Department of Biology, Massachusetts Institute of Technology, Cambridge, United States; [2]Department of Chemistry, Massachusetts Institute of Technology, Cambridge, United States

**Abstract** AAA+ proteases perform regulated protein degradation in all kingdoms of life and consist of a hexameric AAA+ unfoldase/translocase in complex with a self-compartmentalized peptidase. Based on asymmetric features of cryo-EM structures and a sequential hand-over-hand model of substrate translocation, recent publications have proposed that the AAA+ unfoldases ClpA and ClpX rotate with respect to their partner peptidase ClpP to allow function. Here, we test this model by covalently crosslinking ClpA to ClpP to prevent rotation. We find that crosslinked ClpAP complexes unfold, translocate, and degrade protein substrates in vitro, albeit modestly slower than uncrosslinked enzyme controls. Rotation of ClpA with respect to ClpP is therefore not required for ClpAP protease activity, although some flexibility in how the AAA+ ring docks with ClpP may be necessary for optimal function.

*For correspondence:
tabaker@mit.edu

[†]These authors contributed equally to this work

## Introduction

The AAA+ (ATPases Associated with diverse cellular Activities) protease subfamily uses the energy of ATP hydrolysis to disassemble and degrade proteins that are misfolded, deleterious, or unneeded (*Sauer and Baker, 2011*). AAA+ proteases are composed of a hexameric single- or double-ringed AAA+ unfoldase/translocase and a self-compartmentalized partner peptidase. The AAA+ rings form a shallow helix and stack with planar peptidase rings (*Puchades et al., 2020*). After protein substrate recognition by the unfoldase, repeated cycles of ATP hydrolysis power conformational changes in the AAA+ motor, promoting substrate unfolding and processive translocation of the resulting polypeptide into the proteolytic chamber of the peptidase for degradation. Recent structural and biochemical studies have illuminated some aspects of this process, but the molecular nature of the stepwise cycles these proteolytic machines use to carry out mechanical unfolding and translocation of protein substrates is still being actively explored (*Puchades et al., 2020*).

The ClpAP protease consists of the ClpA$_6$ AAA+ unfoldase, a double-ring AAA+ enzyme with two AAA+ modules per subunit, and the tetradecameric ClpP$_{14}$ peptidase, which contains two heptameric rings (*Sauer and Baker, 2011*; *Figure 1A*). Thus, the interface between ClpA and ClpP involves an asymmetric six-to-seven subunit mismatch. The ClpXP protease, composed of the single-ring AAA+ ClpX$_6$ unfoldase and the ClpP$_{14}$ peptidase, also has a six-to-seven mismatch, as do proteasomal AAA+ enzymes. How such mismatches are accommodated structurally and whether the mismatches play important roles in the mechanisms of these ATP-dependent proteases has long been a subject of interest. Recent near-atomic-resolution cryo-EM structures of ClpAP and ClpXP reveal that each unfoldase has six flexible peptidase-binding loops protruding from the bottom face of the AAA+ ring that can interact with ClpP$_{14}$ (*Fei et al., 2020a*; *Fei et al., 2020b*; *Ripstein et al., 2020*; *Lopez et al., 2020*). Part of each loop containing a conserved tripeptide motif (IGL in ClpA;

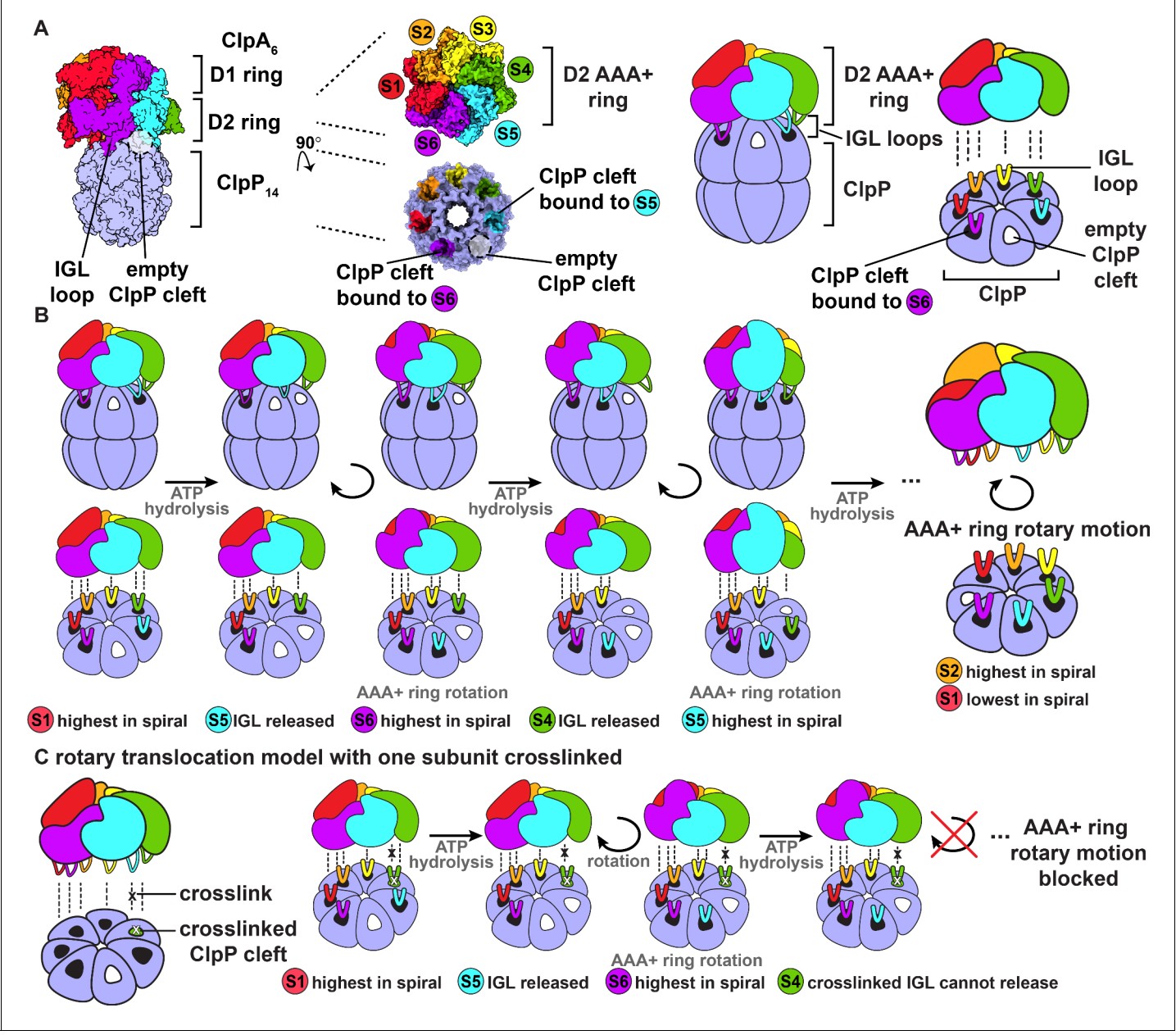

**Figure 1.** ClpAP structure and rotary translocation model. (**A**) Complex of ClpP with ClpA (PDB 6UQO). Subunits of ClpA, labeled S1 through S6, are ordered from the highest to the lowest position in the spiral relative to ClpP at the beginning of the mechanical cycle. The IGL loops of ClpA hexamers dock into a subset of the seven clefts in a heptameric ClpP ring. In this structure, there is an empty cleft between the second lowest and lowest subunits in the spiral (S5 and S6, respectively). The coloring of the ClpP clefts represents the docked position of the IGL loops from the corresponding AAA+ subunits; empty clefts are colored white. The rightmost panel is a generalized model of the ClpA D2 AAA+ ring docking into the ClpP interface. (**B**) Rotary translocation model with clockwise around-the-ring ATP hydrolysis and IGL loop release and rebinding (*Ripstein et al., 2020*; *Lopez et al., 2020*). When subunit S1 is highest in the spiral, ATP hydrolysis releases the IGL loop of subunit S5 and the AAA+ ring rotates clockwise with respect to ClpP. During rotation, subunit S6 moves to the top in of the spiral, and the IGL loop of subunit S5 takes a clockwise 'step' and rebinds to the adjacent empty ClpP cleft. Repetition of this sequence of ATP hydrolysis and IGL loop release and rebinding results in rotary motion of the AAA+ ring with respect to the ClpP ring. (**C**) Rotary translocation model with at least one crosslinked IGL loop. If one ClpA subunit is crosslinked to a ClpP cleft, the rotary motion of the AAA+ ring with respect to the ClpP ring is blocked. The crosslinked ClpA subunit cannot be released from the ClpP cleft and cannot sequentially move to each position in the ClpA spiral.

IGF in ClpX) docks into hydrophobic clefts on the top of the ClpP$_7$ ring, engaging a total of five or six of the seven clefts and leaving one or two clefts unoccupied (*Figure 1A*; *Fei et al., 2020a*; *Fei et al., 2020b*; *Ripstein et al., 2020*; *Lopez et al., 2020*).

Subunits in both the ClpA and ClpX hexamers adopt a shallow helical conformation with axial pore loops that interact with an extended substrate polypeptide to form a structure reminiscent of a spiral staircase. By contrast, ClpP subunits are arranged in near-planar rings that enclose a chamber with luminal peptidase active sites. Other AAA+ proteases have similar architectures, with spiral AAA+ rings and planar peptidase rings (*Puchades et al., 2020*). In structures of heterohexameric AAA+ protease motors in which the positions of unique subunits can be determined, different subunits can occupy the highest and lowest spiral positions, suggesting that dynamic rearrangement of subunits within the spiral is part of the ATP-fueled mechanical cycle that powers substrate translocation (*de la Peña et al., 2018*; *Dong et al., 2019*). In one model for this cycle, an enzyme power stroke is initiated when the second lowest subunit in the spiral hydrolyzes ATP (S5 at the beginning of the cycle, *Figure 1A*), resulting in a rearrangement that moves this subunit and higher subunits, together with bound substrate, each down one position in the spiral, at the same time that the lowest subunit (S6) disengages from substrate and moves to the top of the spiral (*Figure 1B*; *Puchades et al., 2020*). Intriguingly, in recent cryo-EM structures of ClpAP and ClpXP, an empty ClpP cleft is always flanked by clefts that interact with the IGL/IGF loops of the second lowest and lowest subunits within the spiral (S5 and S6 in *Figure 1A*; *Fei et al., 2020a*; *Fei et al., 2020b*; *Ripstein et al., 2020*; *Lopez et al., 2020*). If different subunits in the ClpA or ClpX hexamers pass sequentially through each position in the spiral during substrate translocation and the empty cleft in ClpP is always between clefts that interact with specific subunits in the spiral, then the AAA+ ring should rotate with respect to ClpP during protein translocation (*Figure 1B*; *Ripstein et al., 2020*; *Lopez et al., 2020*).

Here, we test the effects of preventing rotation of the ClpA ring with respect to the ClpP ring by covalently crosslinking multiple IGL loops of ClpA to ClpP (*Figure 1C*). We find that an enzyme containing multiple covalent crosslinks between ClpA and ClpP retains substantial proteolytic activity against unfolded and metastable native substrates but displays defects in degrading more stably folded proteins. We conclude that rotation of ClpA with respect to ClpP is not required for substrate translocation or unfolding, but some freedom of movement at the ClpA-ClpP interface is likely to be important for optimal mechanical activity.

## Results and discussion

### Crosslinking ClpA to ClpP

For crosslinking studies, we used cysteine-free genetic backgrounds for ClpA (C47S, C203S, C243S; *Zuromski et al., 2020*) and ClpP (C91V, C113A; *Amor et al., 2016*). We then introduced an E613C mutation into the IGL loop of otherwise cysteine-free ClpA ($^{E613C}$ClpA$^{‡}$) and appended a cysteine after Asn$^{193}$, the C-terminal residue of otherwise cysteine-free ClpP (ClpP$^{+C}$). Based on cryo-EM structures of ClpAP (*Lopez et al., 2020*), the cysteines introduced by these mutations should be close enough to allow crosslinking of specific subunits of ClpA to neighboring subunits of ClpP. For example, *Figure 2A* shows that Glu$^{613}$ in each subunit of the ClpA hexamer is close to a ClpP Arg$^{192}$ residue, the last ClpP amino acid visible in the ClpAP structure, in six of the seven ClpP protomers. We mixed $^{E613C}$ClpA$^{‡}$ with ClpP$^{+C}$ in the presence of a homobifunctional cysteine crosslinker and then separated covalently joined $^{E613C}$ClpA$^{‡}$–ClpP$^{+C}$ complexes (peak 1) from uncrosslinked ClpP$^{+C}$ (peak 2) by size-exclusion chromatography (*Figure 2B*). After pooling fractions containing $^{E613C}$ClpA$^{‡}$–ClpP$^{+C}$ complexes (*Figure 2B*), quantification by SDS-PAGE revealed that 90 ± 1% of the ClpA was crosslinked to ClpP (designated A–P) (*Figure 2C* lane 7, *Figure 2—source data 1*). Based on this crosslinking efficiency, the vast majority of ClpA hexamers should contain one or more crosslinked A–P subunits (>99.99%, assuming independent crosslinking), and ~98% of hexamers should contain four, five, or six ClpA subunits crosslinked to ClpP (*Figure 2—figure supplement 1*). The $^{E613C}$ClpA$^{‡}$–ClpP$^{+C}$ pool also contained uncrosslinked ClpP$^{+C}$, as expected, and some crosslinked ClpP$^{+C}$ dimers (*Figure 2C*, lane 7).

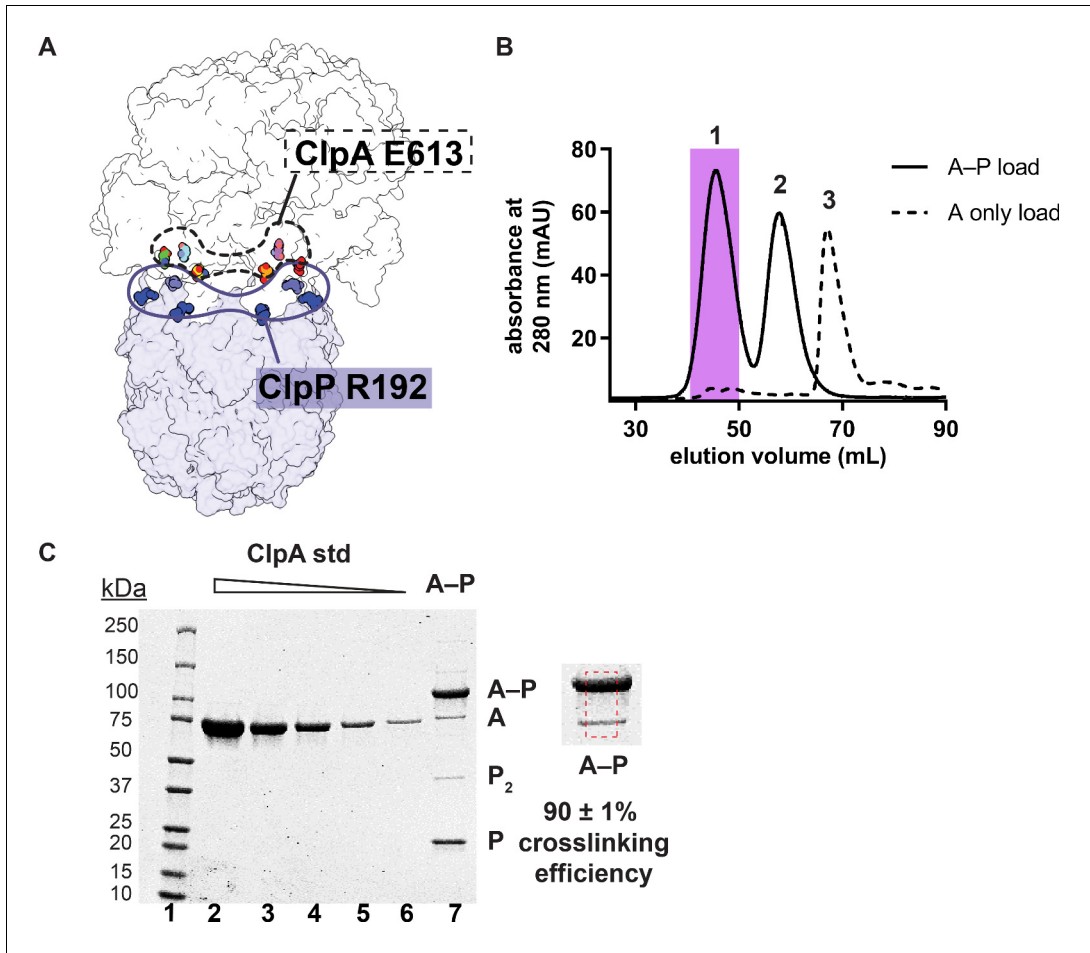

**Figure 2.** ClpA–ClpP crosslinking and purification. (**A**) Proximity of Glu$^{613}$ residues in six subunits of the hexameric ClpA ring (shown in the dashed outline) to Arg$^{192}$ residues in six of the seven subunits of a heptameric ClpP ring (shown in the solid outline). (**B**) Size-exclusion chromatograms of $^{E613C}$ClpA$^{‡}$–ClpP$^{+C}$ following crosslinking (solid line; peaks 1 and 2) or uncrosslinked $^{E613C}$ClpA$^{‡}$ (dashed line; peak 3), which is largely monomeric under the chromatography conditions. As shown in panel C, most ClpA in peak one is crosslinked to ClpP. The shaded area in peak 1 represents the crosslinked A–P that was pooled and used in all experiments in this study. Peak 2 corresponds to uncrosslinked ClpP$^{+C}$ remaining after the crosslinking reaction and chromatographs at the position expected for a tetradecamer. (**C**) Reducing SDS-PAGE of the peak-1 pool. Lanes 1–6 are MW standards or different concentrations of purified $^{E613C}$ClpA$^{‡}$. Lane 7 is an aliquot of the peak-1 pool. The shift to higher molecular weight from uncrosslinked ClpA (A) to crosslinked ClpA–ClpP (A–P) is consistent with covalent linkage of a single ClpA monomer (~83 kDa) to a single ClpP monomer (~23 kDa). The dashed red box is a zoomed-in view of lane 7 used to calculate crosslinking efficiency of $^{E613C}$ClpA$^{‡}$ to ClpP$^{+C}$. Crosslinking efficiency was calculated as the mean ± 1 SD of four independent replicates. The quantification of SDS-PAGE bands used to calculate crosslinking efficiency is available in *Figure 2—source data 1*.

The online version of this article includes the following source data and figure supplement(s) for figure 2:

**Source data 1.** Quantification of A–P crosslinking efficiency.

**Figure supplement 1.** Histogram of the expected number of crosslinks between $^{E613C}$ClpA$^{‡}$ and ClpP$^{+C}$ in the A–P pool assuming independent crosslinking of individual ClpA and ClpP subunits with 90 ± 1% efficiency.

## Crosslinked complexes degrade model substrates

To test if ClpA rotation relative to ClpP is required for ATP-fueled proteolysis, we measured ATP hydrolysis and degradation of model substrates by the purified crosslinked A–P pool *in vitro* compared to an A•P control consisting of assembled but uncrosslinked $^{E613C}$ClpA$^{‡}$ and wild-type ClpP (ClpP$^{WT}$). The A–P pool hydrolyzed ATP at a steady-state rate of 412 ± 40 min$^{-1}$ enz$^{-1}$, whereas this rate was 1035 ± 78 min$^{-1}$ enz$^{-1}$ for the A•P control. The A•P control hydrolyzed ATP and degraded

ssrA-tagged proteins at comparable rates to wild-type ClpAP (ClpA$^{WT}$ClpP$^{WT}$) and a cysteine-free ClpA (ClpA$^{CF}$) variant (ClpA$^{CF}$ClpP$^{WT}$), demonstrating that the C47S, C203S, C243S, and E613C mutations do not impair activity (*Figure 3—figure supplement 1A–B*, *Figure 3—source data 5*). We also observed similar rates of ATP hydrolysis and ssrA-tagged protein degradation when ClpP$^{+C}$, the ClpP variant used for crosslinking, was paired with ClpA$^{WT}$, ClpA$^{CF}$, or $^{E613C}$ClpA$^{\ddagger}$.

To test the effects of crosslinking ClpA to ClpP on proteolysis, we measured the rate at which the A–P and A•P enzymes degraded proteins with a range of native stabilities. These substrates included the N-terminal domain of the phage λ cI repressor with an ssrA tag (λ cI$^N$-ssrA; *Gottesman et al., 1998*), $^{cp7}$GFP-ssrA (*Nager et al., 2011*), $^{5-IAF}$V13P titin$^{I27}$-ssrA (*Kenniston et al., 2003*; *Iosefson et al., 2015*), and FITC-casein (*Twining, 1984*; *Thompson et al., 1994*). Under the conditions of these assays *in vitro*, the A–P sample degraded the folded substrates (λ cI$^N$-ssrA and $^{cp7}$GFP-ssrA) at rates that were 31 ± 5% and 32 ± 4%, respectively, of the A•P control, and degraded unfolded $^{5-IAF}$V13P titin$^{I27}$-ssrA and FITC-casein at 46 ± 2% and 97 ± 7%, respectively, of the control rates (*Figure 3A–B*, *Figure 3—source datas 1–2*). The rate of degradation of FITC-casein by the A–P pool was reduced ~6-fold when ATPγS was substituted for ATP (*Figure 3C*, *Figure 3—source data 3*), indicating that robust degradation of this molten-globule substrate requires ATP hydrolysis. We also determined steady-state kinetic parameters for degradation of $^{cp7}$GFP-ssrA by the A–P pool and A•P control (*Figure 3D*, *Figure 3—source data 4*). Compared to the A•P control, V$_{max}$ was ~50% and $K_M$ was ~3-fold weaker for degradation of this substrate by the A–P pool. This reduction in V$_{max}$ for the A–P pool was roughly comparable to its reduced ATP-hydrolysis activity, suggesting that slower degradation of folded substrates by the A–P pool results from slower ATPase activity. Thus, our results show that multiple crosslinks between ClpA and ClpP in the A–P pool cause modest slowing of the rates of ATP hydrolysis and protein degradation compared to the uncrosslinked controls (*Figure 3—figure supplement 1A–B*), with more prominent degradation defects for native substrates. Notably, however, crosslinks between ClpA and ClpP do not prevent the protein unfolding or translocation steps required for proteolysis. Only the crosslinked A–P sample exhibited substantially lower ATP-hydrolysis and protein degradation activity compared to the uncrosslinked controls; thus, the reduced A–P enzymatic activities are likely to be direct consequences of introducing specific covalent crosslinks between ClpA and ClpP rather than other modifications introduced in the experimental design.

## Mechanistic implications of crosslinked complex activity

Models in which ClpA or ClpX must rotate with respect to ClpP to allow substrate translocation (*Figure 1B–C*) predict that crosslinking ClpA or ClpX to ClpP would abolish protein degradation by stopping rotation and linked sequential movements of ClpA/ClpX subunits through each position in the spiral. Our experimental results do not support these models. Rather, we find that preventing rotation by 'riveting' the ClpA ring to the ClpP ring still permits substantial degradation of native and denatured protein substrates *in vitro*. ClpXP complexes in which one IGF loop is crosslinked to ClpP can also degrade folded and unfolded substrates, albeit at lower rates than uncrosslinked controls (*Bell, 2020*). The high degree of crosslinking in our ClpAP experiments, where 98% of complexes contain at least four covalent crosslinks between ClpA and ClpP, and ~50% of complexes are predicted to contain six crosslinks (*Figure 2—figure supplement 1*), would be expected to hinder each ClpA subunit from adopting each position in the spiral by affecting conformational accessibility, especially near the ClpP interface. Moreover, in approximately half of the crosslinked enzymes, it would not be possible to have two empty ClpP clefts. Hence, the proposal that this intermediate is a requisite step in translocation, as proposed for ClpXP (*Ripstein et al., 2020*), is also inconsistent with our results. *Lopez et al., 2020* proposed that ClpA and ClpP might rotate in defined contexts, for example during the degradation of very stable substrates. Although we cannot exclude this possibility, we prefer simpler models in which the basic mechanism of AAA+ protease function does not change in a substrate-specific manner. As we observe reduced rates of degradation of folded substrates when ClpA and ClpP are crosslinked, conformational flexibility between the unfoldase and protease appears to be important for optimal unfolding. However, rotation of the ClpA or ClpX rings with respect to ClpP is clearly not a strict requirement for degradation. We suggest, therefore, that ring-ring rotation models be considered to be both unproven and unlikely in the absence of direct evidence for such rotation.

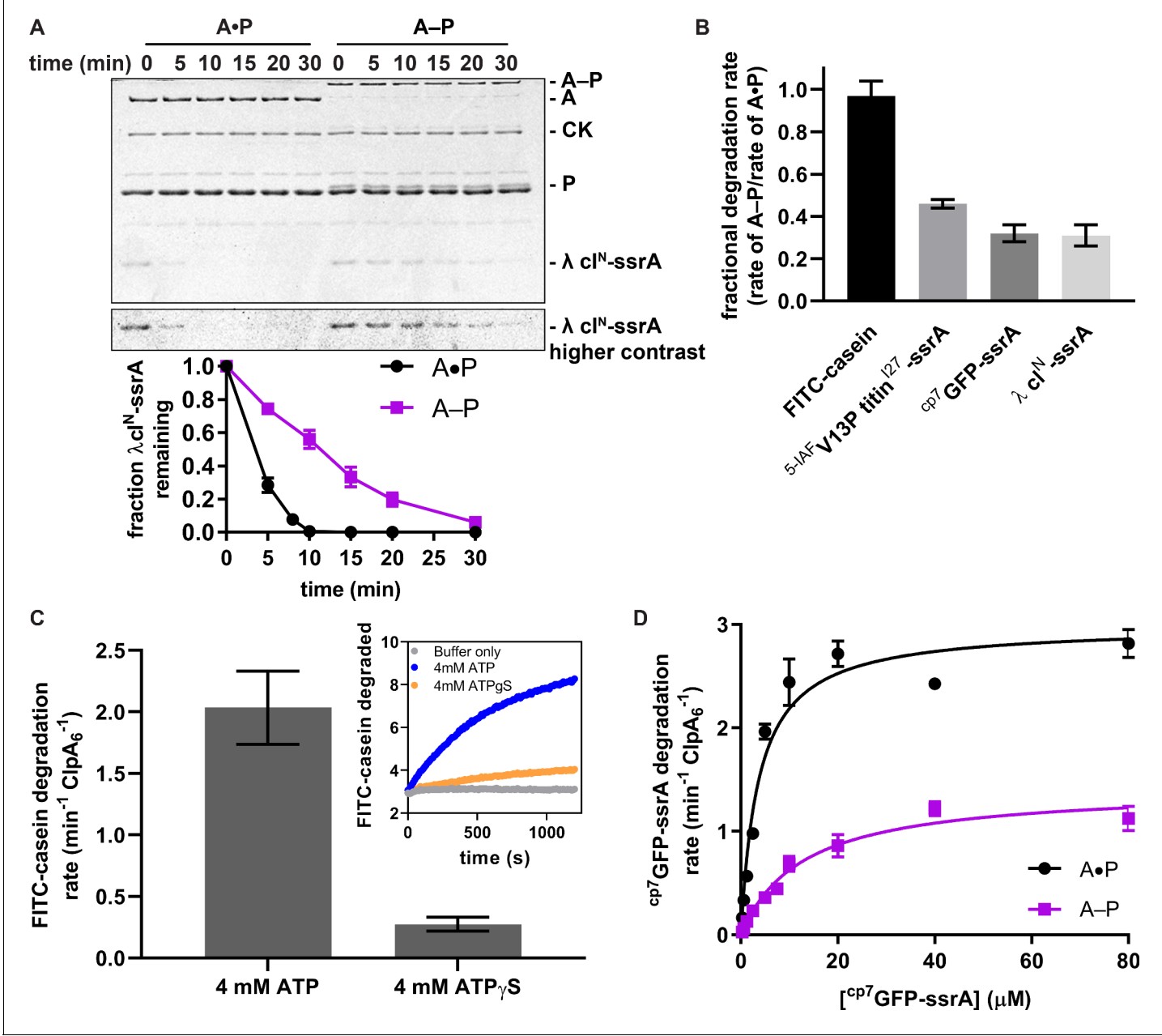

**Figure 3.** Substrate degradation by crosslinked ClpAP (A–P) and uncrosslinked ClpAP (A•P). (**A**) Top, SDS-PAGE assay of the kinetics of λ cl[N]-ssrA degradation by A–P and the A•P control (CK is creatine kinase). Bottom, quantification of λ cl[N]-ssrA degradation. Values are means ± 1 SD (n = 3) and provided in *Figure 3—source data 1*. (**B**) Degradation of substrates of varying thermodynamic stability (18 µM) FITC-casein, 5 µM [5-IAF]V13P titin[I27]-ssrA, 20 µM [cp7]GFP-ssrA, 15 µM λ cl[N]-ssrA by A–P. Fractional degradation rates were calculated by dividing the degradation rates of A–P by the A•P rates. Values are means ± propagated error (n ≥ 3) and provided in *Figure 3—source data 2*. (**C**) Degradation of FITC-casein (18 µM) by A–P in the presence of ATP or ATPγS. FITC-casein degradation was quantified by normalizing the relative fluorescence units to the total FITC-casein degraded upon porcine elastase addition at the endpoint of the assay and subtracting the contributions of photobleaching from the buffer-only control. Values are means ± 1 SD (n = 3) and provided in *Figure 3—source data 3*. The inset shows representative degradation kinetics. (**D**) Michaelis-Menten analysis of [cp7]GFP-ssrA degradation kinetics by A–P and the A•P control. Values are means ±1 SD (n = 3) and provided in in *Figure 3—source data 4*. For A–P degradation, $V_{max}$ was 1.4 ± 0.07 min$^{-1}$ ClpA$_6^{-1}$, $K_M$ was 13 ± 1.6 µM, and $R^2$ was 0.96; for the A•P control, $V_{max}$ was 3.0 ± 0.10 min$^{-1}$ ClpA$_6^{-1}$, $K_M$ was 3.7 ± 0.5 µM, and $R^2$ was 0.96, where the errors are those of non-linear least-squares fitting to the Michaelis-Menten equation.

The online version of this article includes the following source data and figure supplement(s) for figure 3:

**Source data 1.** Quantification of λ cl[N]-ssrA degradation kinetics.
**Source data 2.** Degradation of substrates of varying thermodynamic stability.
**Source data 3.** Degradation of FITC-casein (18 µM) by the purified A–P pool in the presence of ATP or ATPγS.
*Figure 3 continued on next page*

*Figure 3 continued*

**Source data 4.** Michaelis-Menten analysis of $^{cp7}$GFP-ssrA degradation kinetics.

**Source data 5.** ATPase and $^{cp7}$GFP-ssrA degradation rates by ClpAP controls.

**Figure supplement 1.** ATPase and degradation activities of uncrosslinked $^{E613C}$ClpA$^{‡}$ClpP$^{+C}$ and $^{E613C}$ClpA$^{‡}$ClpP$^{WT}$ (A•P) complexes are comparable to those of wild-type ClpAP.

AAA+ proteases in the FtsH/Yme1/Agf3l2 and Lon families have AAA+ and peptidase modules that are genetically tethered as part of the same polypeptide chain and therefore also must operate without rotation between the unfoldase and protease components (*Glynn, 2017*). For ClpAP, a non-rotary mechanism could be explained by the sequential hand-over-hand mechanism if the empty ClpP cleft can localize between any pair of ClpA subunits. Alternative mechanisms, such as the reciprocating action of one or two AAA+ subunits, might also explain both the observed pattern of unfoldase-protein interactions seen in cryo-EM structures and the robust degradation activity of genetically or biochemically tethered AAA+ proteases against multiple substrates. Further experiments will be required to discriminate between these models.

# Materials and methods

## Key resources table

| Reagent type (species) or resource | Designation | Source or reference | Identifiers | Additional information |
|---|---|---|---|---|
| Gene (*Escherichia coli*) | clpA | | | UniProtKB - P0ABH9 |
| Gene (*Escherichia coli*) | clpP | | | UniProtKB - P0A6G7 |
| Strain, strain background (*Escherichia coli*) | T7 Express | New England Biolabs | C2566I | Chemically competent cells |
| Recombinant DNA reagent | pT7 ClpP$^{+C}$ (plasmid) | This paper | | For overexpression of C-terminally His$_6$-tagged ClpP (C91V, C113A) with extra Cys residue for crosslinking. Progenitor: pT7 ClpP-TEV-cHis$_6$ (*Stinson et al., 2013*; *Amor et al., 2016*) |
| Recombinant DNA reagent | pET23b His$_7$Sumo FLAG $^{E613C}$ClpA$^{‡}$ (plasmid) | This paper | | For overexpression of ClpA with Cys substitution and C47S, C203S, C243S background for crosslinking. Progenitor: pET23b His$_7$Sumo ClpA$^{cf}$ΔC9 (*Zuromski et al., 2020*) |
| Recombinant DNA reagent | pET23b His$_7$Sumo ClpA$^{cf}$ΔC9 (plasmid) | *Zuromski et al., 2020* | | For overexpression of cysteine-free ClpA (ClpA$^{CF}$) harbouring C47S, C203S, C243S mutations |
| Recombinant DNA reagent | WT ClpA (plasmid) | *Seol et al., 1994*, *Hou et al., 2008* | | WT ClpA (M169T background) for overexpression |
| Recombinant DNA reagent | ClpP-His$_6$ (plasmid) | *Kim et al., 2001* | | WT ClpP for overexpression |

*Continued on next page*

*Continued*

| Reagent type (species) or resource | Designation | Source or reference | Identifiers | Additional information |
|---|---|---|---|---|
| Recombinant DNA reagent | $^{cp7}$GFP-ssrA (plasmid) | *Nager et al., 2011* | | Circularly permutated variant of superfolder GFP-ssrA for overexpression |
| Recombinant DNA reagent | V13P titin$^{I27}$-ssrA (plasmid) | *Kenniston et al., 2003* | | ssrA-tagged I27 domain variant for overexpression |
| Recombinant DNA reagent | His$_6$SUMO λ cI$^N$-ssrA (plasmid) | This paper | | ssrA-tagged residues 1–93 of λ cI (UniProtKB - P03034) for overexpression |
| Chemical compound, drug | Bismaleimidoethane | Thermo Fisher Scientific | Cat # 22323 | |
| Chemical compound, drug | Adenosine 5′-O-(3-Thiotriphosphate), Tetralithium Salt | Millipore Sigma | Cat# 119120–25 MG | |
| Chemical compound, drug | 5-Iodoacetami dofluorescein | Thermo Fisher Scientific | Cat# I30451 | |
| Chemical compound, drug | Casein fluorescein isothiocyanate from bovine milk (FITC-casein) | Sigma-Aldrich | Cat# C0528-10MG | |

## Proteins

The gene encoding *E. coli* ClpP$^{+C}$ was generated using PCR mutagenesis, and the corresponding protein was purified by established protocols (*Martin et al., 2005*) and stored in buffer containing 0.5 mM dithiothreitol (DTT). Wild-type ClpP (ClpP$^{WT}$) was purified by established protocols (*Kim et al., 2001*). The plasmid for $^{E613C}$ClpA$^{‡}$ was generated by PCR mutagenesis of *E. coli* ClpA$^{ΔC9}$ fused to the 3′-end of His$_7$SumoFLAG cloned into pET23b (Novagen). His$_7$SumoFLAG-$^{E613C}$ClpA$^{‡}$ was overexpressed in T7Express (New England Biolabs), and initially purified by Ni-NTA chromatography. Following ULP-1 cleavage to remove His$_7$Sumo, $^{E613C}$ClpA$^{‡}$ was further purified by SP-Sepharose cation-exchange chromatography and stored in 50 mM HEPES-KOH (pH 7.5), 300 mM NaCl, 20 mM MgCl$_2$, 10% glycerol, and 2 mM TCEP. Cysteine-free (ClpA$^{CF}$) and wild-type (ClpA$^{WT}$) were purified by established protocols (*Zuromski et al., 2020*; *Hou et al., 2008*). The $^{cp7}$GFP-ssrA and V13P titin$^{I27}$-ssrA proteins were purified as described (*Nager et al., 2011*; *Kenniston et al., 2003*). V13P titin$^{I27}$-ssrA was labeled with 5-iodoacetamidofluorescein (5-IAF) for fluorescent assays as described (*Iosefson et al., 2015*). The plasmid for λ cI$^N$-ssrA was generated by PCR mutagenesis of a gene encoding amino acids 1–93 the bacteriophage λ cI repressor. This construct was fused to the 3′-end of His$_6$Sumo cloned into pET23b and appended with the C-terminal ssrA degron. His$_6$Sumo-λ cI$^N$-ssrA was purified by Ni-NTA chromatography, ULP-1 cleavage, Ni-NTA chromatography to remove the His$_6$Sumo fragment, Mono-Q anion-exchange chromatography, and Superdex-75 size-exclusion chromatography, and stored in 25 mM HEPES-KOH (pH 7.5), 150 mM NaCl, 10% glycerol, and 1 mM DTT. ClpA and variant concentrations were calculated as hexamer equivalents, and ClpP and variant concentrations were calculated as tetradecamer equivalents.

## Crosslinking ClpA to ClpP

$^{E613C}$ClpA$^{‡}$ (4 µM) and ClpP$^{+C}$ (9.6 µM) were mixed in a total volume of 2.5 mL and desalted into 50 mM HEPES-KOH (pH 7), 300 mM NaCl, 20 mM MgCl$_2$, 10% glycerol, and 5 mM EDTA using a Sephadex G-25 PD-10 column (GE Healthcare). After diluting to a final volume of 5 mL, crosslinking was initiated by addition of 5 mM ATPγS and 200 µM bismaleimidoethane (BMOE; Thermo Fisher) and allowed to proceed at room temperature for 45 min. The reaction was quenched by addition of 50 mM DTT at room temperature for 20 min before purification by Superdex-200 size-exclusion

chromatography in 50 mM HEPES-KOH (pH 7.5), 300 mM NaCl, 20 mM MgCl₂, 10% glycerol, and 2 mM TCEP. The purified A–P pool was used for all subsequent biochemical assays. Quantification of crosslinking and the concentration of A–P were measured by quantifying Coomassie-stained SDS-PAGE bands relative to $^{E613C}ClpA^{‡}$ standards. The area under the curve (AUC) corresponding to pixel intensities of the crosslinked and uncrosslinked species were quantified by ImageQuant (GE Healthcare) after scanning Coomassie-stained SDS-PAGE using a Typhoon FLA 9500 (GE Healthcare). Crosslinking efficiency was measured in four independent replicates by multiplying the calculated concentration of each species by the volume loaded in each lane, and calculated as:

$$\text{Efficiency} = \frac{\text{picomoles A} - P_{\text{Crosslinked}}}{\text{picomoles A} - P_{\text{Crosslinked}} + \text{picomoles}^{E613C}ClpA^{‡}_{\text{Uncrosslinked}}}$$

## Biochemical assays

We determined the concentration of $^{E613C}ClpA^{‡}$–$ClpP^{+C}$ by a standard-curve comparison to $^{E613C}ClpA^{‡}$ (Figure 2C). We calculated the concentration of the uncrosslinked $^{E613C}ClpA^{‡}$ species by measuring the absorbance at 280 nm (ε = 32890 M$^{-1}$ cm$^{-1}$) using a NanoDrop One UV-Vis Spectrophotometer (Thermo-Fisher Scientific), ATP-hydrolysis assays were performed using an NADH-coupled assay (Martin et al., 2005) at 30°C in Buffer HO (25 mM HEPES-KOH, pH 7.5, 300 mM NaCl, 20 mM MgCl₂, 10% glycerol, 2 mM TCEP) with 5 mM ATP and 0.25 µM $^{E613C}ClpA^{‡}$ and 0.75 µM $ClpP^{WT}$ for the A•P control; 0.25 µM $^{E613C}ClpA^{‡}$–$ClpP^{+C}$ for the A–P pool; or the combinations of 0.25 µM $ClpA^{WT}$, $ClpA^{CF}$, or $^{E613C}ClpA^{‡}$ and 0.75 µM $ClpP^{WT}$ or $ClpP^{+C}$ listed in Figure 3—figure supplement 1A. Degradation reactions were performed at 30°C in Buffer HO with 4 mM ATP and an ATP-regeneration system consisting of 50 µg/mL creatine kinase (Millipore-Sigma) and 5 mM creatine phosphate (Millipore-Sigma). Degradation of $^{cp7}GFP$-ssrA was monitored by loss of substrate fluorescence (excitation 467 nm; emission 511 nm) using a SpectraMax M5 plate reader (Molecular Devices) (Nager et al., 2011). The $^{cp7}GFP$-ssrA concentration was 20 µM in Figure 3B and Figure 3—figure supplement 1B; concentrations varying from 0.31 to 80 µM in Figure 3D contained 0.25 µM $^{E613C}ClpA^{‡}$ and 0.75 µM $ClpP^{WT}$ for the A•P control or 0.25 µM $^{E613C}ClpA^{‡}$–$ClpP^{+C}$ for the A–P pool. In the $^{cp7}GFP$-ssrA degradation assays shown in Figure 3—figure supplement 1B, degradation reactions for uncrosslinked controls included the indicated combinations of 0.25 µM $ClpA^{WT}$, $ClpA^{CF}$, or $^{E613C}ClpA^{‡}$ and 0.75 µM $ClpP^{WT}$ or $ClpP^{+C}$, in addition to 0.25 µM $^{E613C}ClpA^{‡}$ and 0.75 µM $ClpP^{WT}$ for the A•P control or 0.25 µM $^{E613C}ClpA^{‡}$–$ClpP^{+C}$ for the A–P pool. Degradation of FITC-casein (18 µM, Sigma-Aldrich) containing 0.25 µM $^{E613C}ClpA^{‡}$ and 0.75 µM $ClpP^{WT}$ for the A•P control or 0.25 µM $^{E613C}ClpA^{‡}$–$ClpP^{+C}$ for the A–P pool was monitored by increase in fluorescence (excitation 340 nm; emission 520 nm); to determine the endpoint of complete FITC-casein degradation, 0.5 µL of 5 mg/mL porcine elastase (Sigma-Aldrich) was added to each well and incubated for 30 min. ClpAP degradation reactions with FITC-casein (18 µM) were performed at 30°C in Buffer HO with 4 mM ATP or ATPγS (Millipore Sigma). Degradation of $^{5-IAF}V13P$ titin$^{I27}$-ssrA (5 µM) containing 0.2 µM $^{E613C}ClpA^{‡}$ and 0.5 µM $ClpP^{WT}$ for the A•P control or 0.2 µM $^{E613C}ClpA^{‡}$–$ClpP^{+C}$ for the A–P pool was monitored by increase in fluorescence (excitation 494 nm; emission 518 nm). Gel degradation of λ cI$^{N}$-ssrA (15 µM monomer) containing 0.2 µM $^{E613C}ClpA^{‡}$ and 0.4 µM $ClpP^{WT}$ for the A•P control or 0.2 µM $^{E613C}ClpA^{‡}$–$ClpP^{+C}$ for the A–P pool was performed in triplicate by taking samples of each reaction at specific time points, stopped by addition of SDS-PAGE loading sample and boiling at 100°C before loading on Tris-Glycine-SDS gels. Bands were visualized by staining with colloidal Coomassie G-250 and quantified by ImageQuant (GE Healthcare) after scanning by Typhoon FLA 9500 (GE Healthcare). The fraction of λ cI$^{N}$-ssrA remaining was calculated by dividing the intensity of this band at a given time point by the density at time zero, after normalization by the creatine kinase density. The biochemical assays were performed with A–P from a single preparation to ensure that crosslinking efficiency was the same throughout all assays. All experiments were performed in at least three independent replicates and values reported were calculated as the mean ±1 SD of independent replicates or the ratio of means ± propagated error of independent replicates. Propagated error for 'fractional degradation rate (rate of A–P/rate of A•P)' of A–P mean activity compared to A•P mean activity was computed as:

$$\text{Propagated error of } \frac{\text{mean}_{A-P}}{\text{mean}_{A\bullet P}} = \frac{\text{mean}_{A-P}}{\text{mean}_{A\bullet P}}\sqrt{\left(\frac{\text{SD}_{A-P}}{\text{mean}_{A-P}}\right)^2 + \left(\frac{\text{SD}_{A\bullet P}}{\text{mean}_{A\bullet P}}\right)^2}$$

## Acknowledgements

We are grateful to X Fei (MIT) for helpful advice on preparation of crosslinked samples and A Torres-Delgado (MIT) and HC Kotamarthi (MIT) for providing materials. We thank JR Kardon, AO Olivares, and KR Schmitz for thoughtful comments on the manuscript. This work was supported by NIH grants GM-101988 (RTS), AI-016892 (RTS, TAB), and the Howard Hughes Medical Institute (TAB). TA Bell and SK were supported by NIH training grant 5T32GM-007287; SK and KLZ were supported by NSF grant GRFP 1745302.

## Additional information

### Funding

| Funder | Grant reference number | Author |
| --- | --- | --- |
| National Institute of General Medical Sciences | GM-101988 | Robert T Sauer |
| National Institute of Allergy and Infectious Diseases | AI-016892 | Robert T Sauer<br>Tania A Baker |
| National Institute of General Medical Sciences | 5T32GM-007287 | Sora Kim<br>Kristin L Zuromski<br>Tristan A Bell |
| National Science Foundation | GRFP 1745302 | Sora Kim<br>Kristin L Zuromski |
| Howard Hughes Medical Institute | | Tania A Baker |

The funders had no role in study design, data collection and interpretation, or the decision to submit the work for publication.

### Author contributions

Sora Kim, Kristin L Zuromski, Conceptualization, Resources, Formal analysis, Funding acquisition, Validation, Investigation, Visualization, Methodology, Writing - original draft, Project administration, Writing - review and editing; Tristan A Bell, Conceptualization, Resources, Formal analysis, Validation, Investigation, Methodology, Writing - original draft, Writing - review and editing; Robert T Sauer, Conceptualization, Resources, Formal analysis, Supervision, Funding acquisition, Validation, Methodology, Writing - original draft, Project administration, Writing - review and editing; Tania A Baker, Conceptualization, Resources, Supervision, Funding acquisition, Validation, Methodology, Writing - original draft, Project administration, Writing - review and editing

### Author ORCIDs

Sora Kim https://orcid.org/0000-0002-9856-9574
Kristin L Zuromski https://orcid.org/0000-0003-0960-5009
Tristan A Bell https://orcid.org/0000-0002-3668-8412
Robert T Sauer http://orcid.org/0000-0002-1719-5399
Tania A Baker https://orcid.org/0000-0002-0737-3411

### Decision letter and Author response

Decision letter https://doi.org/10.7554/eLife.61451.sa1
Author response https://doi.org/10.7554/eLife.61451.sa2

## Additional files

### Supplementary files
- Transparent reporting form

### Data availability
All data generated or analysed during this study are included in the manuscript and supporting files. Source data files have been provided for Figures 2 and 3.

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
