## [Decision Letter]

**Acceptance summary:**

Your finding that cross-linked ClpA-ClpP retains some proteolytic activity (albeit substantially diminished for some substrates) argues against recent proposals that substrate processing is coupled to ClpA-ClpP rotation. This is an important clarification for the fast-moving field of AAA protease mechanisms.

**Decision letter after peer review:**

Thank you for submitting your article "ClpAP proteolysis does not require rotation of the ClpA unfoldase relative to ClpP" for consideration by *eLife*. Your article has been reviewed by three peer reviewers, including Christopher P Hill as the Reviewing Editor and Reviewer #1, and the evaluation has been overseen by Cynthia Wolberger as the Senior Editor.

The reviewers have discussed the reviews with one another and the Reviewing Editor has drafted this decision to help you prepare a revised submission.

Summary:

In principle, this Short Report might be suitable for publication in *ELife* because it uses a crosslinking approach to establish that the paradigmatic AAA ATPase ClpA does not have to rotate with respect to its ClpP protease partner during substrate translocation. This is important because the mechanism of AAA ATPases is of considerable interest and recent high-profile publications have inferred that ClpA and its relative ClpX do rotate with respect to ClpP during translocation.

Essential revisions:

1) The model being tested is built on the observation of Lopez et al. that in cryo-EM structures the single empty cleft of ClpP is always located between the clefts occupied by the IGL loops of the lowest and second lowest ClpA subunits. (And on an equivalent observation and proposal for ClpX.) Is it possible that other conformations actually exist, but are less stable and therefore missed in the cryo-EM analysis? If so, one would not have to postulate that ClpA (or ClpX) and ClpP rotate relative to one another, but the hand-over-hand model for how substrate is moved by the ClpA subunits would still be valid.

2) The Discussion includes "the proposal that this intermediate is a requisite step in translocation (Lopez et al., 2020) is inconsistent with our results". This seems like an overstatement of the Lopez et al. wording "this rotation may be substrate specific and perhaps more critical for the proteolysis of stable, folded substrates compared to labile structures". Indeed, the Lopez et al. wording seems consistent with the current manuscript finding that degradation of folded substrates is impacted to a greater extent by ClpAP crosslinking than unfolded substrates.

3) Direct comparison of the ATPase and protease activities of the ClpA cysteine triple mutant and the ClpP double mutant should be made with WT proteins in the absence of cross-linking.

4) The paper hinges on the demonstration that the protein tested for degradation contains more than 90% of crosslinked A-P complex. However, it is not entirely clear how the experiments were done. In the legend to Figure 2C, it says that "a fraction of the A-P pool" was analyzed. Do the authors mean that they analyzed an aliquot of the pool or a fraction of the gel filtration run? Were the enzymatic reactions in Figure 3 all done with the same A-P pool? How was the variation of crossslinking efficiency estimated (+/-1% seems a bit too accurate)? From different experiments with different ClpA and ClpP preps? Are the error bars in Figure 3 derived from experiments with different preps or from repetitions with a single A-P pool?

5) To demonstrate that equivalent concentrations of non-crosslinked and crosslinked complex were used in the experiments in Figure 3, a Coomassie-stained gel of the two complexes should be shown.

6) The quantity being shown in Figure 3B should be clarified. Is it the ratio of degradation rates, or is it a comparison of end point values, etc?

---

## [Author Response]

Essential revisions:1) The model being tested is built on the observation of Lopez et al. that in cryo-EM structures the single empty cleft of ClpP is always located between the clefts occupied by the IGL loops of the lowest and second lowest ClpA subunits. (And on an equivalent observation and proposal for ClpX.) Is it possible that other conformations actually exist, but are less stable and therefore missed in the cryo-EM analysis? If so, one would not have to postulate that ClpA (or ClpX) and ClpP rotate relative to one another, but the hand-over-hand model for how substrate is moved by the ClpA subunits would still be valid.

The reviewers are correct to point out that more conformations of the ClpA-ClpP interface may exist but are not represented in the Lopez *et al.* cryo-EM structures. This issue brings up the more general point that, especially if the available structural datasets and analyses are incomplete, the general features of the hand-over-hand model for translocation and the proposal that the ClpA and ClpP enzymes rotate as a consequence of translation do not need to be coupled features of the same mechanism. A corollary of this statement therefore is that although we do not observe enzyme-enzyme rotation as a critical aspect of the reaction cycle, our finding that ClpA rotation relative to ClpP is not obligatory for function does not exclude the “hand-over-hand” model for polypeptide translocation by AAA+ unfoldases.

Importantly, the model for ClpAP function that is specifically excluded by our results is a hand-over-hand model coupled with continuous cycling of the unfoldase loops to the structurally observed positions (postulated by Ripstein *et al*., 2020 and Lopez *et al*., 2020). We have now expanded on how these two features of model(s) for the enzyme cycle are separable under specific scenarios.

2) The Discussion includes "the proposal that this intermediate is a requisite step in translocation (Lopez et al., 2020) is inconsistent with our results". This seems like an overstatement of the Lopez et al. wording "this rotation may be substrate specific and perhaps more critical for the proteolysis of stable, folded substrates compared to labile structures". Indeed, the Lopez et al. wording seems consistent with the current manuscript finding that degradation of folded substrates is impacted to a greater extent by ClpAP crosslinking than unfolded substrates.

We appreciate the point that we misstated the claims in Lopez et al., 2020, (which were included in their bioRxiv preprint but not in the final NSMB paper). We have revised this portion of the Discussion to include a more nuanced description of how our findings can be interpreted in light of the report of Lopez and colleagues. Rather than the possibility that AAA+ proteases, such as ClpAP, utilize different enzymatic mechanisms depending on the identity of the substrate (e.g., rotation of ClpA with respect to ClpP is substrate-specific and is greater utilized for folded substrates than unfolded ones), we attribute constrained conformational accessibility owing to the multiple crosslinks between ClpA and ClpP (which may hinder ClpA from adopting each position in the AAA+ spiral), to explain the greater impact of crosslinking on degradation rates of folded compared to unfolded substrates. We agree with the reviewers that more experiments are needed to understand the conformational changes in ClpAP that promote and accompany its mechanical operations and have tried to make this issue clear in the Discussion.

3) Direct comparison of the ATPase and protease activities of the ClpA cysteine triple mutant and the ClpP double mutant should be made with WT proteins in the absence of cross-linking.

As requested, we now include experiments comparing the ATP-hydrolysis and protease activities of ClpA and ClpP variants used for the crosslinking studies. As suggested by the reviewer, these variants are compared to the wild-type ClpAP enzyme in the absence of crosslinking. Furthermore, the ^E613C^ClpA^‡^ and ClpP^+C^ enzymes were assayed with and without chemical crosslinking (see new Figure 3—figure supplement 1). We also now show biochemical characterizations of cysteine-free ClpA (ClpA^CF^) with both wild-type ClpP (ClpP^WT^) and ClpP^+C^. Biochemical characterizations of ClpA^CF^ and cysteine-free ClpP have been described previously (Zuromski *et al*., 2020; Amor *et al*., 2016) and are referenced in the paper.

Although the ATP-hydrolysis rates show more variation than the protein-degradation rates, it is clear that all of the ClpA/ClpP combinations are highly functional, within ~70% of the value of the wild-type control. Thus, the removal/introduction of Cys residues needed for the crosslinking experiments did not substantially affect ClpAP activity. Importantly, uncrosslinked ^E613C^ClpA^‡^ in combination with ClpP^+C^ had rates of ATP hydrolysis and ^cp7^GFP-ssrA degradation equivalent to wild-type ClpAP. Indeed, the only enzyme pair exhibiting substantially lower activity was crosslinked ^E613C^ClpA^‡^—ClpP^+C^. Thus, we conclude that the changes in activity that we report are a consequence of the introduction of specific crosslinks between ClpA and ClpP.

4) The paper hinges on the demonstration that the protein tested for degradation contains more than 90% of crosslinked A-P complex. However, it is not entirely clear how the experiments were done. In the legend to Figure 2C, it says that "a fraction of the A-P pool" was analyzed. Do the authors mean that they analyzed an aliquot of the pool or a fraction of the gel filtration run? Were the enzymatic reactions in Figure 3 all done with the same A-P pool? How was the variation of crossslinking efficiency estimated (+/-1% seems a bit too accurate)? From different experiments with different ClpA and ClpP preps? Are the error bars in Figure 3 derived from experiments with different preps or from repetitions with a single A-P pool?

We apologize to the reviewers for the lack of clarity in our description in Figure 2C and have revised the figure legend in Figure 2 accordingly. In Figure 2C, we present analysis of an aliquot of the purified and pooled crosslinked A–P (peak 1; after the size-exclusion chromatography step). All enzymatic reactions in Figure 3 were performed with samples from this same pool of purified A–P. The crosslinking efficiency was also determined using this purified A–P preparation.

Our purification procedures as well as a more complete description of measuring crosslinking efficiency are included in the updated Materials and method*s* section. Briefly, the procedure was as follows: four samples from the A–P master pool were run on SDS-PAGE, and the concentration of free ClpA (A) and the crosslinked (A–P) species were determined by staining and densitometry. This analysis was repeated with four samples and used to calculate the mean value for the crosslinked species ± 1 SD.

5) To demonstrate that equivalent concentrations of non-crosslinked and crosslinked complex were used in the experiments in Figure 3, a Coomassie-stained gel of the two complexes should be shown.

As requested by the reviewers, we have updated Figure 3A to include the image of the entire stained gel that was used to compare the degradation time courses for A•P and A–P. In each case, the A or the A–P crosslinked protein are the largest species on the gel, and can now be viewed directly by the reviewers/readers. (Unfortunately, the A–P species migrates near the top of the gel). We attribute the slightly weaker density and therefore smaller amount of crosslinked A–P enzyme compared to ClpA in the uncrosslinked A•P sample to differences in protein concentration determination methods of A–P compared to A•P. We measured the concentration of A–P by staining and densitometry of A–P bands relative to ^E613C^ClpA^‡^ standards, whereas in the uncrosslinked A•P control sample, concentrations of ^E613C^ClpA^‡^ and ClpP^WT^ were determined by Nanodrop measured at the absorbance at 280 nm (ε for ClpA = 32,890 M^-1^ cm^-1^; ε for ClpP = 8,940 M^-1^ cm^-1^). We estimate that the measurements of the two protein concentration methods to be within ± 10% of each other.

6) The quantity being shown in Figure 3B should be clarified. Is it the ratio of degradation rates, or is it a comparison of end point values, etc?

The y-axis value in Figure 3B is now replaced with the wording “fractional degradation rate (rate of A–P/ rate of A•P)” instead of “fraction crosslinked activity” to clarify that this value is a ratio of degradation rates. The figure legend has also been updated to further clarify this point.